# Scalable Non-Equivariant 3D Molecule Generation via Rotational Alignment

**Yuhui Ding**[1]  **Thomas Hofmann**[1]

## Abstract

Equivariant diffusion models have achieved impressive performance in 3D molecule generation. These models incorporate Euclidean symmetries of 3D molecules by utilizing an SE(3)-equivariant denoising network. However, specialized equivariant architectures limit the scalability and efficiency of diffusion models. In this paper, we propose an approach that relaxes such equivariance constraints. Specifically, our approach learns a sample-dependent SO(3) transformation for each molecule to construct an aligned latent space. A non-equivariant diffusion model is then trained over the aligned representations. Experimental results demonstrate that our approach performs significantly better than previously reported non-equivariant models. It yields sample quality comparable to state-of-the-art equivariant diffusion models and offers improved training and sampling efficiency. Our code is available at https://github.com/skeletondyh/RADM.

## 1. Introduction

Diffusion-based generative models (Ho et al., 2020; Song et al., 2021) have achieved rapid and significant progress in recent years. These models feature a forward process and a reverse process. The forward process is a fixed Markov process that gradually adds noise to each data sample until the original information is totally corrupted. The reverse process is parameterized by a denoising neural network that is trained to reverse the forward process step by step and recover the original data from noise. Once trained, new samples can be generated by performing the reverse process starting from non-informative noise. Motivated by the success of diffusion models in vision tasks (Dhariwal & Nichol, 2021; Rombach et al., 2022), researchers have sought to extend them to broader areas beyond vision, such as natural language (von Rütte et al., 2025), molecules (Hoogeboom et al., 2022), proteins (Yim et al., 2023).

In this paper, we focus on the task of 3D molecule generation, which has been an important and active topic in drug discovery. Specifically, our goal is to generate the 3D atomic coordinates and atom types of a molecule from scratch. Different from data with grid-like structures (e.g., images and text sequences), 3D molecules pose unique challenges to generative modeling due to the Euclidean symmetry group of $\mathbb{R}^3$, i.e., SE(3). Particularly, the chemical properties of a molecule remain unchanged no matter how the molecule is translated or rotated in the three-dimensional space. To incorporate the Euclidean symmetries into the model, Hoogeboom et al. (2022) proposed an equivariant diffusion model EDM that aims to learn an SO(3)-invariant distribution, which requires the score function of the diffusion model to be SO(3)-equivariant (Köhler et al., 2020; Garcia Satorras et al., 2021). To satisfy such a constraint, EDM utilized an EGNN (Satorras et al., 2021) to parameterize the denoising network. The impressive performance of EDM on the 3D molecule generation task has inspired a series of follow-up works (Wu et al., 2022; Xu et al., 2023; Vignac et al., 2023b) that build equivariance into diffusion models. Consequently, equivariant diffusion models have become the dominant approach for tasks related to 3D molecules.

While equivariant diffusion models account for the symmetries of 3D molecules, an important question arises naturally: *Is equivariance necessary for an effective molecular generative model?* We argue that it is not, as the probability of a molecule is determined by the total probability of all its possible 3D positions, regardless of whether each position has an equal probability. Furthermore, equivariant architectures have drawbacks. Compared with their non-equivariant counterparts, they have more complex parametrization and lack standardized implementations. Additionally, equivariant architectures tend to be less efficient and less scalable than non-equivariant ones. In contrast to the vision and text domains which have been unified by transformers (Vaswani et al., 2017; Dosovitskiy et al., 2021), irregular data from scientific domains, with equivariant models being the mainstream choice, have benefited less from recent advances in architectural optimization (Dao et al., 2022; Peebles & Xie, 2023).

[1]Department of Computer Science, ETH Zurich. Correspondence to: Yuhui Ding <yuhui.ding@inf.ethz.ch>.

*Proceedings of the 42$^{nd}$ International Conference on Machine Learning*, Vancouver, Canada. PMLR 267, 2025. Copyright 2025 by the author(s).

Driven by the interest in further investigating the capacity of non-equivariant diffusion models and motivated by the possibility of connecting advances in different domains, in this paper we propose an approach to improving non-equivariant diffusion models for 3D molecule generation. Our inspiration comes from 3D vision models that typically rely on clean and well-aligned datasets such as ShapeNet (Chang et al., 2015). We hypothesize that aligning the representations of 3D molecules could mitigate the variation in 3D coordinates caused by arbitrary and unknown Euclidean transformations, thereby making the data distribution more learnable for a non-equivariant generative model.

However, aligning 3D molecules is inherently challenging due to the absence of supervision signals. To address this challenge, we propose to construct aligned representations in an unsupervised manner with an autoencoder. Autoencoders have been widely used to reduce input dimensions and improve efficiency for latent diffusion models (Rombach et al., 2022). In our approach, the autoencoder shapes the latent space that is more suitable for non-equivariant diffusion models. More specifically, we use a separate graph neural network to generate a sample-dependent SO(3) transformation (in the form of a rotation matrix) to rotate each molecule, and then train a non-equivariant autoencoder to reconstruct the rotated molecules. The network that generates the rotation is trained jointly with the autoencoder to minimize the reconstruction error, so that the model learns to arrange molecules in a way that facilitates the learning of a non-equivariant autoencoder. The aligned latent space then allows us to train a non-equivariant diffusion model for which a non-equivariant architecture (such as a vanilla graph neural network or a transformer (Peebles & Xie, 2023)) can be employed as the denoising network, offering improved efficiency and better scalability. We refer to our approach as RADM (Rotationally Aligned Diffusion Model). We empirically test our approach on molecule generation benchmarks, and experimental results demonstrate that our approach significantly outperforms previously reported non-equivariant models and achieves sample quality comparable to state-of-the-art equivariant diffusion models. As expected, our non-equivariant diffusion model exhibits better scalability and improves the sampling efficiency significantly.

## 2. Preliminaries

We review the necessary background in this section before we present our proposed approach. In Section 2.1, we formally describe the definition of the problem. In Section 2.2, we introduce the basics of diffusion models. To simplify the expressions, we use the notion of signal-to-noise ratio following Kingma et al. (2021). In Section 2.3, we summarize how existing works incorporate equivariance into the diffusion model, upon which we build our own approach.

### 2.1. Problem Definition

We are interested in generating 3D molecules from scratch. We consider each molecule as a collection of points in the three dimensional space. Specifically, a molecule that has $N$ atoms is represented as $\boldsymbol{x} = (\boldsymbol{x}_1, \boldsymbol{x}_2, \ldots, \boldsymbol{x}_N) \in \mathbb{R}^{N \times 3}$, which correspond to the atomic coordinates in $\mathbb{R}^3$, and $\boldsymbol{h} = (\boldsymbol{h}_1, \boldsymbol{h}_2, \ldots, \boldsymbol{h}_N) \in \mathbb{R}^{N \times d}$, which correspond to the features of atoms (e.g., types and charges). The atomic coordinates $\boldsymbol{x}$ are affected by translations and rotations in $\mathbb{R}^3$, while the atom features $\boldsymbol{h}$ are invariant to such transformations. Let $\mathbf{R} \in \mathbb{R}^{3 \times 3}$ be an orthogonal rotation matrix and $\boldsymbol{t} \in \mathbb{R}^3$, the transformed coordinates can be written as[1]:

$$\mathbf{R}\boldsymbol{x} + \boldsymbol{t} = (\mathbf{R}\boldsymbol{x}_1 + \boldsymbol{t}, \mathbf{R}\boldsymbol{x}_2 + \boldsymbol{t}, \ldots, \mathbf{R}\boldsymbol{x}_N + \boldsymbol{t}). \quad (1)$$

### 2.2. Diffusion Models

The forward process of the diffusion model starts with a generic data point $\boldsymbol{x}$ and adds increasing levels of Gaussian noise to it. We use $\boldsymbol{z}_t$ to denote the noisy version of $\boldsymbol{x}$ for $t = 0, 1, \ldots, T$:

$$q(\boldsymbol{z}_t | \boldsymbol{x}) = \mathcal{N}(\boldsymbol{z}_t | \alpha_t \boldsymbol{x}, \sigma_t^2 \boldsymbol{I}) \quad (2)$$

where $\alpha_t$ controls how much information of the original data point $\boldsymbol{x}$ is retained and $\sigma_t$ defines the level of added noise. In this paper we follow previous works (Ho et al., 2020; Kingma et al., 2021) and let $\alpha_t^2 + \sigma_t^2 = 1$. $\alpha_t$ is close to 1 at $t = 0$ and then monotonically decreases to $\alpha_T$ that is close to 0, which means the original data point is gradually corrupted until it becomes almost pure noise. The forward process is defined to be Markovian and the joint distribution of all noisy variables can be written as:

$$q(\boldsymbol{z}_0, \boldsymbol{z}_1, \ldots, \boldsymbol{z}_T | \boldsymbol{x}) = q(\boldsymbol{z}_0 | \boldsymbol{x}) \prod_{t=1}^{T} q(\boldsymbol{z}_t | \boldsymbol{z}_{t-1}) \quad (3)$$

Then the process (2) can be equivalently expressed using a transition distribution $q(\boldsymbol{z}_t | \boldsymbol{z}_s)$ for $0 \leq s < t \leq T$:

$$q(\boldsymbol{z}_t | \boldsymbol{z}_s) = \mathcal{N}(\boldsymbol{z}_t | \alpha_{t|s} \boldsymbol{z}_s, \sigma_{t|s}^2 \boldsymbol{I}) \quad (4)$$

where $\alpha_{t|s} = \alpha_t / \alpha_s$ and $\sigma_{t|s}^2 = \sigma_t^2 - \alpha_{t|s}^2 \sigma_s^2$. Given the distributions above and using Bayes' rule, we can derive the true posterior distribution of the forward transition (4), conditioned on $\boldsymbol{x}$:

$$q(\boldsymbol{z}_s | \boldsymbol{z}_t, \boldsymbol{x}) = \mathcal{N}(\boldsymbol{z}_s | \boldsymbol{\mu}_{s|t}(\boldsymbol{z}_t, \boldsymbol{x}), \sigma_{s|t}^2 \boldsymbol{I}) \quad (5)$$

where $\boldsymbol{\mu}_{s|t}(\boldsymbol{z}_t, \boldsymbol{x})$ and $\sigma_{s|t}^2$ can be analytically computed:

$$\boldsymbol{\mu}_{s|t}(\boldsymbol{z}_t, \boldsymbol{x}) = \frac{\alpha_{t|s} \sigma_s^2}{\sigma_t^2} \boldsymbol{z}_t + \frac{\alpha_s \sigma_{t|s}^2}{\sigma_t^2} \boldsymbol{x}, \quad \sigma_{s|t}^2 = \frac{\sigma_{t|s}^2 \sigma_s^2}{\sigma_t^2} \quad (6)$$

---

[1]Here $\mathbf{R}\boldsymbol{x} + \boldsymbol{t}$ denotes the group action of $(\mathbf{R}, \boldsymbol{t})$ on $\boldsymbol{x}$.

The reverse process of diffusion models is also Markovian. It aims to generate all intermediate variables $z_t$ and $x$ in the reverse direction:

$$p(z_0, z_1, \ldots, z_T, x) = p(z_T)p(x|z_0) \prod_{t=1}^{T} p(z_{t-1}|z_t) \quad (7)$$

where $p(z_T)$ is chosen to be a standard normal distribution. Since the exact reverse transition $p(z_s|z_t)$ $(0 \le s < t \le T)$ is unknown, it is parameterized by a neural network $\phi$ that is trained to approximate the true posterior conditioned on $x$ (5):

$$p_\phi(z_s|z_t) = q(z_s|z_t, x_\phi(z_t, t)) \quad (8)$$

In (8) the network $\phi$ is expected to predict the original data point $x$ from its noisy version $z_t$. Ho et al. (2020) found that predicting the noise added to $x$ yielded better performance than predicting $x$ itself. We use the same noise prediction parametrization in our model, and $x_\phi(z_t, t)$ in (8) is further rewritten as:

$$x_\phi(z_t, t) = \frac{z_t}{\alpha_t} - \frac{\sigma_t}{\alpha_t} \epsilon_\phi(z_t, t) \quad (9)$$

Using similar techniques as VAEs, the evidence lower bound (for a single data point $x$) can be derived for the diffusion model as:

$$-\log p(x) \le -\text{ELBO}(x) = \mathcal{L}_{\text{prior}} + \mathcal{L}_0 + \sum_{t=1}^{T} \mathcal{L}_t \quad (10)$$

where $\mathcal{L}_{\text{prior}} = D_{\text{KL}}\left(q(z_T|x)\|p(z_T)\right)$ denotes the KL divergence between the final distribution $q(z_T|x)$ generated by the forward process and the prior distribution $p(z_T)$, $\mathcal{L}_0 = \mathbb{E}_{q(z_0|x)}\left[-\log p(x|z_0)\right]$ represents a reconstruction loss, and $\mathcal{L}_t = \mathbb{E}_{q(z_t|x)}D_{\text{KL}}(q(z_{t-1}|z_t, x)\|p(z_{t-1}|z_t))$. Let $\text{SNR}(t)$ be the signal-to-noise ratio (Kingma et al., 2021) defined as $\text{SNR}(t) = \alpha_t^2/\sigma_t^2$, the term $\mathcal{L}_t$ can be further expanded (using the noise parametrization) as:

$$\mathcal{L}_t = \mathbb{E}_{\epsilon \sim \mathcal{N}(0, I)} \left[ \frac{1}{2} \left( \frac{\text{SNR}(t-1)}{\text{SNR}(t)} - 1 \right) \left\| \epsilon - \epsilon_\phi \right\|^2 \right] \quad (11)$$

where $\epsilon_\phi$ denotes $\epsilon_\phi(\alpha_t x + \sigma_t \epsilon, t)$, the predicted noise that is output by the denoising network $\phi$.

To obtain the training objective, we follow the practice of DDPM (Ho et al., 2020) and discard the weighting in (11). Note that $\mathcal{L}_{\text{prior}}$ is a constant irrelevant of optimization and $p(x|z_0)$ can be parameterized by a separate Gaussian distribution to make $\mathcal{L}_0$ have a similar form to (11), the final training objective is:

$$\mathcal{L}(x) = \mathbb{E}_{\epsilon \sim \mathcal{N}(0, I), t \sim \mathcal{U}\{0, 1, \ldots, T\}} \left[ \left\| \epsilon - \epsilon_\phi \right\|^2 \right] \quad (12)$$

where $t$ is uniformly sampled between 0 and $T$.

## 2.3. Equivariance

As mentioned in Section 2.1, the atomic coordinates $x$ of a molecule can be arbitrarily translated and rotated in the three-dimensional space without affecting its chemical properties, which poses challenges to generative modeling. The translation transformation can be handled trivially by projecting the coordinates of the molecule into the $(N-1) \times 3$ dimensional linear subspace where $\sum_i x_i = 0$. The projection can be made by subtracting the center of gravity from each $x_i$. As proven by Garcia Satorras et al. (2021) and Xu et al. (2022), sampling from a normal distribution in the $(N-1) \times 3$ dimensional subspace can be done by first sampling from a normal distribution in the $N \times 3$ dimensional space and then subtracting the center of gravity. The diffusion model in the subspace can then be derived in the same way as in Section 2.2, except that the center of gravity needs to be subtracted from the part of the predicted noise that corresponds to atomic coordinates, to restrict intermediate variables to the subspace.

To deal with rotations, most existing works (Hoogeboom et al., 2022; Xu et al., 2023) add equivariance constraints to the diffusion model. Specifically, they model a distribution that is invariant to rotation $\mathbf{R}$:

$$p(x) = p(\mathbf{R}x) \quad (13)$$

Köhler et al. (2020) showed that applying an equivariant invertible transformation to an invariant distribution would result in another invariant distribution. Formally, a function $f$ is SO(3) equivariant if it satisfies $f(\mathbf{R}x) = \mathbf{R}f(x)$ for any $\mathbf{R} \in$ SO(3). Xu et al. (2022) further showed that for a Markov chain with an invariant initial distribution, if the Markov transition kernel is equivariant:

$$p(x'|x) = p(\mathbf{R}x'|\mathbf{R}x), \quad (14)$$

then the marginal distribution at any time step is invariant. In the context of diffusion models, the starting distribution of the generative process $p(z_T)$ is isotropic Gaussian that naturally satisfies (13). To ensure that the final distribution defined by the entire generative process is invariant, the transition density function $p(z_s|z_t)$ (8) needs to be equivariant, which requires the noise prediction network $\phi$ (9) to be an equivariant neural network. EGNN (Satorras et al., 2021) has been the most popular architecture for $\phi$. One EGNN layer that takes $x^l, h^l$ as inputs and outputs $x^{l+1}, h^{l+1}$ is defined as:

$$m_{ij} = \phi_e(h_i^l, h_j^l, d_{ij}^2, a_{ij}),$$
$$h_i^{l+1} = \phi_h(h_i^l, \sum_{j \ne i} \tilde{e}_{ij} m_{ij}),$$
$$x_i^{l+1} = x_i^l + \sum_{j \ne i} \frac{x_i^l - x_j^l}{d_{ij} + 1} \phi_x(h_i^l, h_j^l, d_{ij}^2, a_{ij}), \quad (15)$$

where $d_{ij} = \|x_i^l - x_j^l\|_2$ denotes the Euclidean distance, $a_{ij}$ is optional edge attribute and $\tilde{e}_{ij} = \phi_{\inf}(m_{ij})$ reweighs messages coming from different atoms. Learnable parameters are composed of $\phi_e, \phi_h, \phi_x, \phi_{\inf}$, which are all implemented as MLPs.

## 3. Method

While equivariant diffusion models have become the mainstream choice for 3D molecule generation, the equivariance constraints (13) and (14) may not be necessary for a good 3D molecule generator. In this section, we propose a method to relax such constraints and make non-equivariant diffusion models work comparably well. We are inspired by 3D vision models that typically benefit from well-aligned datasets such as ShapeNet (Chang et al., 2015) and thus want to construct aligned representations for molecules. In Section 3.1, we introduce how we learn alignment with an autoencoder. In Section 3.2, we describe the non-equivariant diffusion models that run in the aligned latent space.

### 3.1. Aligned Latent Space

In this section, we aim to address the following two key questions: (1) *How should we represent an alignment operation?* and (2) *How can we learn an effective alignment for each molecule?* Following the practice introduced in Section 2.3, we deal with translations by subtracting the center of gravity for each molecule and restricting the diffusion model to the subspace. Then, the variation in atomic coordinates can only be caused by rotation $\mathbf{R} \in SO(3)$. Therefore, aligning a molecule can be thought of as rotating it into a particular orientation. There are multiple ways to represent a rotation (e.g., Euler angles, exponential coordinates), but not all of them are good for gradient-based learning (Brégier, 2021; Geist et al., 2024). Based on recent analysis (Geist et al., 2024), we choose to parameterize a rotation with an arbitrary matrix $M \in \mathbb{R}^{3 \times 3}$ which is then projected to SO(3) using the singular value decomposition (SVD):

$$\mathbf{R} = \text{SVD}^+(M) = U \text{diag}\left(1, 1, \det(UV^\top)\right) V^\top \quad (16)$$

where $U, V \in \mathbb{R}^{3 \times 3}$ are obtained by SVD: $M = U\Sigma V^\top$, and $\det\left(UV^\top\right)$ ensures that $\det(\mathbf{R}) = 1$.

The above rotation representation is good for gradient-based optimization in the sense that $\text{SVD}^+(M)$ is smooth where $\det(M) \neq 0$ (Levinson et al., 2020). Brégier (2021) provided an efficient toolbox[2] that contains this method and is well compatible with PyTorch's automatic differentiation. Furthermore, there is no restriction on the input $M$, making it suitable for building on top of a neural network. To generate a sample-dependent rotation for each molecule, we

[2] https://github.com/naver/roma

---

**Algorithm 1** Training algorithm for the autoencoder.

**Inputs:** atomic coordinates $x$, atom features $h$
**Learnable parameters:** rotation network $\mathcal{R}_\theta$, encoder $\mathcal{E}_\eta$, decoder $\mathcal{D}_\psi$
**while** not converged **do**
  $\mathbf{R}_\theta \leftarrow \mathcal{R}_\theta(x, h)$
  $\mu_x, \mu_h \leftarrow \mathcal{E}_\eta(\mathbf{R}_\theta x, h)$
  Subtract center of gravity from $\mu_x$
  $\epsilon = (\epsilon_x, \epsilon_h) \sim \mathcal{N}(0, I)$
  Subtract center of gravity from $\epsilon_x$
  $z_x, z_h \leftarrow \mu + \sigma\epsilon$
  Calculate the reconstruction loss $\mathcal{L}(\theta, \eta, \psi)$ (20)
  Update $\theta, \eta, \psi$
**end while**

---

let $M$ be the output of a vanilla GNN (i.e., the rotation network). Let $u = [x, h]$ denote the concatenation of atomic coordinates and features, a GNN layer gives:

$$m_{ij} = \phi_e(u_i^l, u_j^l), \quad u_i^{l+1} = \phi_u\left(u_i^l, \sum_{j \neq i} \tilde{e}_{ij} m_{ij}\right) \quad (17)$$

where $\phi_e$ and $\phi_u$ are MLPs and $\tilde{e}_{ij}$ is parameterized similarly to (15). Comparing (17) to (15), the only difference is that the vanilla GNN layer discards the equivariant update of atomic coordinates and is thus non-equivariant, but it is much simpler. We take the average of all atom representations output by the last layer and feed it into a 2-layer head to obtain $M$.

With a properly parameterized rotation representation, the next challenge is how to learn good sample-dependent rotations. This is non-trivial since there is no direct supervision signal. To overcome this, we propose to learn rotations in an unsupervised manner through an autoencoder. The intuition is that well-aligned molecules should facilitate the learning of a non-equivariant autoencoder. We note that existing work GeoLDM (Xu et al., 2023) introduced a latent diffusion model for 3D molecule generation. However, both the encoder and the decoder used by GeoLDM are equivariant (parametrized by EGNNs (15)). Therefore, the relative orientations between different molecules cannot be adjusted in the latent space, and their latent diffusion model is also an equivariant one. Different from them, our approach can add more flexibility to the latent space, making it more suitable for non-equivariant diffusion models.

Formally, we use $\theta$ to denote the parameters of the GNN that generates rotation representations and use $\mathbf{R}_\theta = \mathbf{R}_\theta(x, h)$ to denote the rotation matrix generated for the molecule $(x, h)$. Then the molecule after applying $\mathbf{R}_\theta$ is denoted as $(\mathbf{R}_\theta x, h)$. Let $\mathcal{E}_\eta$ denote the encoder parameterized by $\eta$ and $\mathcal{D}_\psi$ denote the decoder parameterized by $\psi$, then the

encoding and decoding processes are described as:

$$q_{\theta,\eta}(\boldsymbol{z}_x, \boldsymbol{z}_h | \boldsymbol{x}, \boldsymbol{h}) = \mathcal{N}\left(\mathcal{E}_\eta\left(\mathbf{R}_\theta \boldsymbol{x}, \boldsymbol{h}\right), \sigma^2 \boldsymbol{I}\right) \quad (18)$$

$$p_\psi(\mathbf{R}_\theta \boldsymbol{x}, \boldsymbol{h} | \boldsymbol{z}_x, \boldsymbol{z}_h) = p_\psi(\mathbf{R}_\theta \boldsymbol{x}, \boldsymbol{h} | \mathcal{D}_\psi(\boldsymbol{z}_x, \boldsymbol{z}_h)) \quad (19)$$

where $\boldsymbol{z}_x \in \mathbb{R}^{N \times 3}$, $\boldsymbol{z}_h \in \mathbb{R}^{N \times d'}$ denote the latent representations for $\boldsymbol{x}$ and $\boldsymbol{h}$ respectively. Note that both the input to the encoder $\mathcal{E}$ and the target of the reconstruction are the rotated molecule $(\mathbf{R}_\theta \boldsymbol{x}, \boldsymbol{h})$. In (18) we parameterize the joint distribution of $\boldsymbol{z}_x, \boldsymbol{z}_h$ as an isotropic Gaussian with a fixed variance $\sigma^2$, which allows $q_{\theta,\eta}(\boldsymbol{z}_x, \boldsymbol{z}_h | \boldsymbol{x}, \boldsymbol{h})$ to be decomposed into the product of two Gaussians: $q_{\theta,\eta}(\boldsymbol{z}_x | \boldsymbol{x}, \boldsymbol{h})$ and $q_{\theta,\eta}(\boldsymbol{z}_h | \boldsymbol{x}, \boldsymbol{h})$. As mentioned in Section 2.3, we restrict the distribution $q_{\theta,\eta}(\boldsymbol{z}_x | \boldsymbol{x}, \boldsymbol{h})$ to the subspace where the center of gravity is zero. The reconstruction distribution (19) also admits a factorization: $p_\psi(\mathbf{R}_\theta \boldsymbol{x} | \boldsymbol{z}_x, \boldsymbol{z}_h) p_\psi(\boldsymbol{h} | \boldsymbol{z}_x, \boldsymbol{z}_h)$, with $p_\psi(\mathbf{R}_\theta \boldsymbol{x} | \boldsymbol{z}_x, \boldsymbol{z}_h)$ being a Gaussian in the zero-center-of-gravity subspace and $p_\psi(\boldsymbol{h} | \boldsymbol{z}_x, \boldsymbol{z}_h)$ being a categorical distribution if $\boldsymbol{h}$ is in the form of one-hot encoding for atom types. The reconstruction loss $\mathcal{L}(\theta, \eta, \psi)$ is defined as:

$$\mathcal{L} = -\mathbb{E}_{q_{\theta,\eta}(\boldsymbol{z}_x, \boldsymbol{z}_h | \boldsymbol{x}, \boldsymbol{h})} \left[\log p_\psi(\mathbf{R}_\theta \boldsymbol{x}, \boldsymbol{h} | \boldsymbol{z}_x, \boldsymbol{z}_h)\right] \quad (20)$$

We write down the entire training algorithm for our autoencoder in Algorithm 1. Following Xu et al. (2023), we adopt an early stopping training strategy as the regularization for the encoder.

Regarding the specific architectural choices for $\mathcal{E}_\eta$ and $\mathcal{D}_\psi$, we use the same encoder architecture as GeoLDM (Xu et al., 2023) for the purpose of ablation study, which is a 1-layer EGNN (15). As for the decoder, we use a non-equivariant GNN (17) with the same number of layers as the GeoLDM decoder. The non-equivariance of the decoder is necessary to make the autoencoder sensitive to rotational transformations, thus enabling the learning of aligned representations[3].

### 3.2. Non-Equivariant Latent Diffusion Model

With the aligned latent space, we hypothesize that non-equivariant diffusion models can learn the data distribution more easily. The noise prediction network of a non-equivariant diffusion model has a more flexible architectural design space, since permutation-equivariance is now the only inductive bias of the architecture. A straightforward option is a non-equivariant GNN (17) that simply concatenates atomic coordinates and features as the new atom features and discards the equivariant update of EGNN. The time step $t$ is treated as a scalar feature and is appended to the feature vector of every atom. Another promising

---

[3] Let $\hat{\boldsymbol{x}} = \mathcal{D}(\mathcal{E}(\boldsymbol{x}))$ denote the reconstructed coordinates. The reconstruction loss reduces to L2 loss $\|\hat{\boldsymbol{x}} - \boldsymbol{x}\|_2^2$. If $\mathcal{E}$ and $\mathcal{D}$ were both equivariant, the L2 loss would be invariant to any rotation of $\boldsymbol{x}$ (since rotation matrices are orthogonal), making the rotation network unable to receive any supervision signal.

choice is a transformer model since the attention mechanism is naturally permutation-equivariant. We experiment with the diffusion transformer (DiT) proposed by Peebles & Xie (2023). The DiT essentially follows the standard transformer architecture (Vaswani et al., 2017) and injects conditional information (e.g., time steps) through adaptive normalization parameters output by a 2-layer MLP that takes as input the conditional information. For a more detailed illustration of the DiT architecture, we refer readers to the original paper (Peebles & Xie, 2023). To adapt DiT to our setting, we remove its patch embedding module and covariance prediction head, and add attention masks to deal with varying atom numbers of different molecules. Since the latent representation $\boldsymbol{z}_x$ (18) already encodes the positional information of atoms, we don't apply positional encoding.

The autoencoder and the latent diffusion model are trained separately. We first train the autoencoder following Algorithm 1 and then train the latent diffusion model with the loss function (12), over the latent representations that are sampled according to (18) from the fixed encoder. To generate a molecule, we first sample a molecule size $N$ from the categorical distribution $p(N)$ of molecule sizes on the training set, and then run the reverse process of the latent diffusion model with $N$ fixed. The output of the final step of the reverse process is decoded back to the original space.

## 4. Experiments

In this section, we present experimental results that empirically validate the effectiveness of our proposed approach. In Section 4.1, we introduce the experimental setup, including the datasets, baselines and implementation details. In Section 4.2 and 4.3, we present the main results on molecule generation benchmarks. In Section 4.4, we show the results of ablation studies. In Section 4.5, we demonstrate the efficiency and scalability of our non-equivariant model. Finally in Section 4.6, we show the conditional generation results.

### 4.1. Experimental Setup

**Datasets** We first evaluate our approach using the QM9 dataset (Ramakrishnan et al., 2014) which is a standard molecule generation benchmark widely used by related works. QM9 contains 130K molecules with up to 9 heavy atoms (29 atoms including hydrogens). Each molecule has 3D coordinates, atom types (H, C, N, O, F) and (integer-valued) charges as atom features. We split the dataset in the same way as Hoogeboom et al. (2022), with 100K, 18K, 13K samples for the train, validation and test partitions respectively. Next we evaluate our model on the larger GEOM-Drugs dataset (Axelrod & Gomez-Bombarelli, 2022). This dataset contains 430K molecules with up to 181 atoms and 44.4 atoms on average. Following the setup of Hoogeboom et al. (2022), for each molecule we select the 30 lowest

Table 1: Results of atom stability, molecule stability, validity and validity × uniqueness. Higher numbers are better. We use distinct colors to indicate the best equivariant baseline, the best non-equivariant baseline and our proposed approach. Baseline results are copied from respective papers.

| | QM9 | | | | GEOM-Drugs | |
| --- | --- | --- | --- | --- | --- | --- |
| | Atom Sta (%) | Mol Sta (%) | Valid (%) | Valid & Unique (%) | Atom Sta (%) | Valid (%) |
| Data | 99.0 | 95.2 | 97.7 | 97.7 | 86.5 | 99.9 |
| ENF | 85.0 | 4.9 | 40.2 | 39.4 | - | - |
| G-SchNet | 95.7 | 68.1 | 85.5 | 80.3 | - | - |
| EDM | 98.7 | 82.0 | 91.9 | 90.7 | 81.3 | 92.6 |
| EDM-bridge | 98.8 | 84.6 | 92.0 | 90.7 | 82.4 | 92.8 |
| GeoLDM | **98.9** | **89.4** | **93.8** | **92.7** | **84.4** | **99.3** |
| GDM | 97.0 | 63.2 | - | - | 75.0 | 90.8 |
| GDM-aug | 97.6 | 71.6 | 90.4 | 89.5 | 77.7 | 91.8 |
| GraphLDM | 97.2 | 70.5 | 83.6 | 82.7 | 76.2 | 97.2 |
| GraphLDM-aug | **97.9** | **78.7** | **90.5** | **89.5** | **79.6** | **98.0** |
| RADM$_{\text{DiT-S}}$ | 98.2±0.1 | 83.4±0.2 | 92.5±0.3 | 90.6±0.3 | 83.8 | 98.9 |
| RADM$_{\text{DiT-B}}$ | **98.5±0.0** | **87.3±0.2** | **94.1±0.1** | **91.7±0.1** | **85.0** | **99.3** |

energy conformations. For both datasets, our model learns to generate the 3D coordinates and atom types of complete molecules with explicit hydrogens.

**Baselines** We compare our model with several representative equivariant generative models for 3D molecules, including Equivariant Normalizing Flows (ENF) (Garcia Satorras et al., 2021), G-SchNet (Gebauer et al., 2019), EDM (Hoogeboom et al., 2022), EDM-bridge (Wu et al., 2022) and GeoLDM (Xu et al., 2023). Among them, EDM and GeoLDM are state-of-the-art equivariant diffusion models. Besides, we compare with EDM's non-equivariant counterpart GDM, which uses a non-equivariant GNN (17) as the denoising network, and GDM-aug, which randomly rotates each molecule during training as data augmentation. Similarly, we compare with GeoLDM's non-equivariant versions GraphLDM and GraphLDM-aug.

**Implementation Details** Our implementation is based on the open-source code base of EDM[4] and GeoLDM[5] to keep a fair comparison. We use the same hidden dimension and number of layers for the autoencoder as GeoLDM, and the number of layers of the rotation network is 2 on both datasets. As for the diffusion model, we use the same noise schedule and number of time steps as EDM/GeoLDM. To implement DiT as the noise prediction network, we follow the official code base[6] and make necessary modifications as explained in Section 3.2. We train the autoencoder (and the rotation network) using the Adam optimizer with a learning rate of $1 \times 10^{-4}$ and a cosine annealing schedule. The latent diffusion model is also trained using Adam with a learning rate of $1 \times 10^{-4}$.

---

[4]https://github.com/ehoogeboom/e3_diffusion_for_molecules
[5]https://github.com/MinkaiXu/GeoLDM
[6]https://github.com/facebookresearch/DiT

### 4.2. QM9

We train our model to generate molecules unconditionally. Following Hoogeboom et al. (2022), we use the distance between each pair of atoms and the atom types to decide the bond type (single, double, triple, or none). We don't use any chemical software (e.g., Open Babel) to edit the generated molecules. We report atom stability (the proportion of atoms with the correct valence) and molecule stability (the proportion of molecules for which all atoms are stable) to represent sample quality. We also report validity and uniqueness of the generated molecules (as measured by RDKit).

On QM9, we train the autoencoder for 200 epochs using a batch size of 64. Then we evaluate two variants of the non-equivariant diffusion model based on the same trained autoencoder: RADM$_{\text{DiT-S}}$ and RADM$_{\text{DiT-B}}$. RADM$_{\text{DiT-S}}$ uses the small version DiT (defined in the DiT paper (Peebles & Xie, 2023)) as the noise prediction network, while RADM$_{\text{DiT-B}}$ uses the base version. Compared with DiT-S, DiT-B has twice the hidden size and twice the number of heads. We adopt a batch size of 256 as used in the DiT paper, and train both RADM$_{\text{DiT-S}}$ and RADM$_{\text{DiT-B}}$ for around 5500 epochs.

The results are shown in Table 1. We report the average performance and standard deviation across three runs, each sampling 10000 molecules. For ease of illustration, we categorize baselines into equivariant models and non-equivariant models, and use distinct colors to indicate the best model of each group. As we can see from the table, diffusion models perform much better than ENF and G-SchNet, and equivariant baselines significantly outperform non-equivariant baselines. Notably, both RADM$_{\text{DiT-S}}$ and RADM$_{\text{DiT-B}}$ improve the performance drastically compared with previous

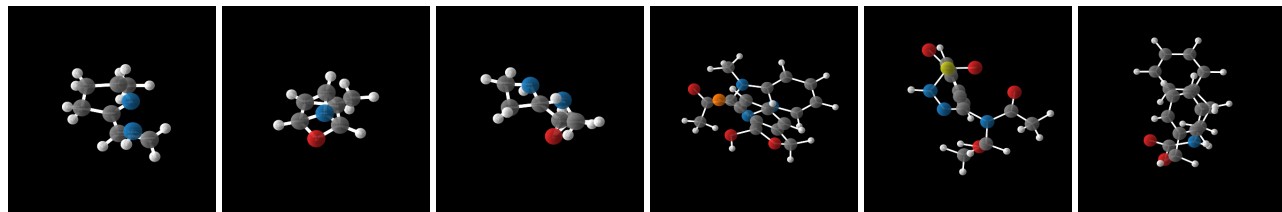

Figure 1: Molecules generated by RADM$_{\text{DiT-B}}$ on QM9 (the three on the left) and GEOM-Drugs (the three on the right).

non-equivariant models. RADM$_{\text{DiT-S}}$ outperforms EDM on molecule stability and outperforms both EDM and EDM-bridge on validity. We find that scaling from DiT-S to DiT-B consistently boosts performance on all metrics. RADM$_{\text{DiT-B}}$ greatly reduces the gap with the best equivariant model. It performs slightly worse than GeoLDM on molecule stability, but achieves the best validity. We note that most hyper-parameters used in our experiments simply follow the equivariant baselines (GeoLDM and EDM), and more tuning may further improve the performance of our non-equivariant model. We visualize the learned rotations in Appendix A.

### 4.3. GEOM-Drugs

On GEOM-Drugs, the autoencoder is trained for 4 epochs using a batch size of 32. Then we train RADM$_{\text{DiT-S}}$ and RADM$_{\text{DiT-B}}$ for around 55 epochs with a batch size of 256. Following previous works, we report the atom stability and validity in Table 1. The performance is averaged over three runs, each sampling 10000 molecules (the standard deviation is negligible after rounding). From the table we observe that RADM exhibits very strong performance on this much larger dataset with more complex molecules. RADM$_{\text{DiT-S}}$ is already highly competitive compared with the best equivariant model GeoLDM. After scaling to DiT-B, RADM$_{\text{DiT-B}}$ outperforms GeoLDM on atom stability while obtaining the same validity.

Table 2: Results (as percentages) of the ablation study on QM9 with the same diffusion backbone.

|  | Atom Sta | Mol Sta | Valid |
|---|---|---|---|
| GraphLDM | 97.2 | 70.5 | 83.6 |
| GraphLDM-aug | 97.9 | 78.7 | 90.5 |
| RADM$_{\text{GNN}}$ (PCA) | 98.1 | 81.9 | 91.7 |
| RADM$_{\text{GNN}}$ | **98.6** | **85.8** | **93.8** |

### 4.4. Ablation Study

To validate the effectiveness of the alignment itself, we conduct an ablation study using the same architecture for the diffusion model. Specifically, we use a basic non-equivariant GNN (17) as the noise prediction network and train the diffusion model based on the same trained autoencoder as in

Section 4.2. We refer to this variant as RADM$_{\text{GNN}}$. The non-equivariant baselines GraphLDM and GraphLDM-aug used the same GNN architecture as the noise prediction network, but were trained in a latent space without learned alignment. We also experiment with a pre-processing method that applies PCA to atomic coordinates and calculates new coordinates by treating the principal components as the new basis. The same diffusion model is then trained on molecules with new coordinates, which we name RADM$_{\text{GNN}}$ (PCA). We train the diffusion models for 3000 epochs using a batch size of 64. The results are shown in Table 2. We can see that RADM$_{\text{GNN}}$ (PCA) outperforms GraphLDM and GraphLDM-aug, which supports the benefit of alignment. However, PCA ignores the dependencies between atomic coordinates and atom types that should be generated together, and the signs of the principal components are ambiguous. The performance of RADM$_{\text{GNN}}$ (PCA) is worse than RADM$_{\text{GNN}}$, which further supports our approach that learns alignment. Interestingly, we also find that RADM$_{\text{GNN}}$ performs better than RADM$_{\text{DiT-S}}$. This is not very surprising since the GNN (17) also has a global receptive field and RADM$_{\text{GNN}}$ is trained with more gradient updates (the batch size is smaller).

Table 3: Comparison of model size, average training time per epoch and sampling efficiency.

|  | #Params | Training | Sampling |
|---|---|---|---|
| EDM | 5.3M | 107.9s | 55s / 100 samples |
| GeoLDM | 11.4M | 118.2s | 49s / 100 samples |
| RADM$_{\text{GNN}}$ | 8.7M | 79.6s | 24s / 100 samples |
| RADM$_{\text{DiT-S}}$ | 37.6M | 38.8s | 7s / 100 samples |
| RADM$_{\text{DiT-B}}$ | 134M | 68.9s | 19s / 100 samples |

### 4.5. Efficiency Comparison

In this section, we compare the training and sampling efficiency of EDM, GeoLDM and RADM. We list the parameter counts, average training time per epoch and sampling speed in Table 3. All the numbers are measured on a single RTX 4090 GPU. We add the number of parameters of the autoencoder but exclude its training time, since training the autoencoder takes only a small portion of the time required to train the latent diffusion model. The sampling speed is measured by the average time used to generate 100 samples

as one mini-batch, and we use 1000 sampling steps for all models. From the table we can see that in general our non-equivariant model RADM is significantly more efficient than EDM and GeoLDM. While RADM$_\text{DiT-S}$ and RADM$_\text{DiT-B}$ have larger model sizes, due to the parallelizable transformer architecture, they remain highly efficient and offer further improved sampling speed. These results demonstrate that our non-equivariant model has better scalability and great potential for further improvement.

Table 4: Mean Absolute Error for molecular property prediction by a pretrained predictor (lower is better). The baseline performance is borrowed from EDM and GeoLDM.

| Property | $\alpha$ | $\Delta\varepsilon$ | $\varepsilon_\text{HOMO}$ | $\varepsilon_\text{LUMO}$ | $\mu$ | $C_v$ |
| Unit | Bohr$^3$ | meV | meV | meV | D | $\frac{\text{cal}}{\text{mol}}$K |
| --- | --- | --- | --- | --- | --- | --- |
| QM9 | 0.10 | 64 | 39 | 36 | 0.043 | 0.040 |
| Random | 9.01 | 1470 | 645 | 1457 | 1.616 | 6.857 |
| EDM | 2.76 | 655 | 356 | 584 | 1.111 | 1.101 |
| GeoLDM | 2.37 | 587 | 340 | 522 | 1.108 | 1.025 |
| RADM$_\text{DiT-B}$ | **1.98** | **458** | **290** | **383** | **0.814** | **0.869** |

### 4.6. Conditional Generation

In this section, we evaluate the ability of our model to generate molecules with desired properties. To assess the quality of the generated molecules regarding the target property, we follow the protocol of EDM (Hoogeboom et al., 2022). The QM9 training set is equally divided into two parts. An EGNN is trained on the first half as a property predictor, while the generative model is trained on the other half. The property predictor is then evaluated on the samples drawn from the generative model. We report the MAE by the pretrained predictor as the evaluation metric in Table 4. The numbers for "QM9" are obtained by evaluating the property predictor directly on the second half of the training set, which can be considered as lower bounds for the model's performance, while "Random" represents a worst-case baseline that evaluates the property predictor using randomly shuffled property labels. We test RADM$_\text{DiT-B}$ on this task. Specifically, our conditional model uses the same pretrained autoencoder as in Section 4.2, but trains the noise prediction network of the latent diffusion model conditioned on the target properties which are appended to each atom's latent representation. From Table 4, we observe that RADM$_\text{DiT-B}$ consistently outperforms both EDM and GeoLDM, further proving the effectiveness of our proposed approach.

## 5. Related Work

### 5.1. Diffusion Models

Diffusion models were first invented in the context of thermodynamics (Sohl-Dickstein et al., 2015). DDPM (Ho et al., 2020) established a connection between diffusion models and denoising score matching and derived a simple training objective via parameterizing a noise prediction model. Song et al. (2021) unified DDPM and score matching with Langevin dynamics (Song & Ermon, 2019) under a general framework defined through continuous stochastic differential equations. Concurrently, Kingma et al. (2021) derived a continuous-time variational lower bound for diffusion models by considering infinitely deep VAEs and simplified the expression in terms of the signal-to-noise ratio. Latent diffusion models (Rombach et al., 2022) reduced computational requirements by training on latent representations output by autoencoders. In addition to reducing the input dimension for diffusion models, autoencoders allow a more flexible latent space which could benefit the training of diffusion models. In contrast to most previous works that used a convolutional U-net as the backbone for the denoising network, Peebles & Xie (2023) proposed to replace the U-net with a transformer and obtained better performance while being more compute-efficient.

### 5.2. Molecule Generation

Molecule generation has always been a central topic in drug discovery. The relevant literature can be categorized according to target tasks. 2D graph generative models (Jin et al., 2020; Jo et al., 2022; Vignac et al., 2023a) only consider node types and edge types, and permutation equivariance is the symmetry of interest. Among them, Jin et al. (2020) trained a VAE to extend a molecular graph in an autoregressive way and used structural motifs as the building blocks. Jo et al. (2022) directly applied continuous-time diffusion models (Song et al., 2021) to node features and adjacency matrices, and Vignac et al. (2023a) utilized discrete diffusion models (Austin et al., 2021). Compared with the 2D structure, 3D coordinates of a molecule contain richer information about its biological activities, so the generation of 3D molecules has attracted increasing interest. To deal with the Euclidean transformations (i.e., translations and rotations) that can affect the 3D coordinates arbitrarily, EDM (Hoogeboom et al., 2022) used an EGNN (Satorras et al., 2021) as the noise prediction network of a diffusion model to ensure an invariant target probability distribution. Wu et al. (2022) enhanced EDM by designing physics informed prior bridges. GeoLDM (Xu et al., 2023) utilized latent diffusion models for 3D molecule generation. Both the autoencoder and the diffusion model of GeoLDM are still equivariant. Different from them, we demonstrate that equivariance is not necessary for a good molecule diffusion model. Concurrently, Zhang et al. (2024) and Joshi et al. (2025) also consider DiT as the backbone of the molecule diffusion model and learn stochastic equivariance through data augmentation. Our paper provides a different perspective that improves non-equivariant diffusion models even without data augmentation. Another line of related work is conformation generation (Xu et al., 2022; Wang et al., 2024) where the 3D conformation of a molecule is predicted

from its 2D structure. Notably, Wang et al. (2024) observed that a non-equivariant model performed better than equivariant baselines. However, in their setting the 2D structure is given during both training and sampling. There are other works (Vignac et al., 2023b; Le et al., 2024) that model 2D and 3D information jointly. These models have access to the 2D structures during training, which is different from our setting.

### 5.3. Learned Canonicalization

Our work is also related to learned canonicalization (Kaba et al., 2023; Dym et al., 2024). In our approach, we don't force the aligned form of a molecule to be exactly rotation-invariant, and more importantly, we learn the alignment in an unsupervised manner. A concurrent work (Sareen et al., 2025) merges an equivariant canonicalization network with a non-equivariant denoising network of a diffusion model and therefore inherits the inefficiency of training equivariant diffusion models. In contrast, our method learns alignment using a lightweight non-equivariant autoencoder and then trains fully non-equivariant diffusion models in the aligned latent space.

## 6. Conclusion

In this paper, we introduce a novel approach that significantly boosts non-equivariant diffusion models on the 3D molecule generation task, through constructing a rotationally aligned latent space. Our approach performs comparably to state-of-the-art equivariant diffusion models, with favorable scalability and improved efficiency.

## Impact Statement

This paper presents work whose goal is to advance the field of Machine Learning. There are many potential societal consequences of our work, none of which we feel must be specifically highlighted here.

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

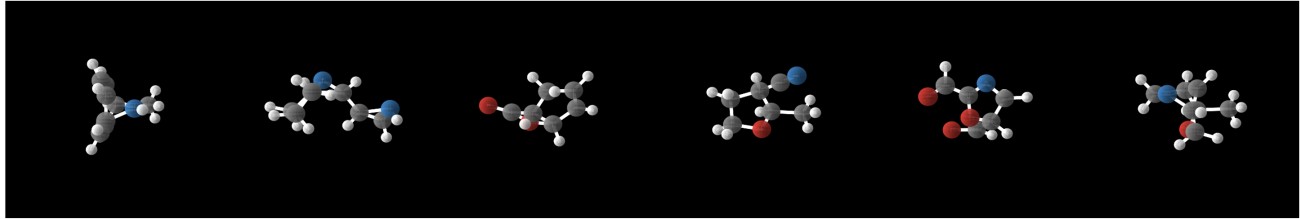

Figure 2: Molecule samples from the training set.

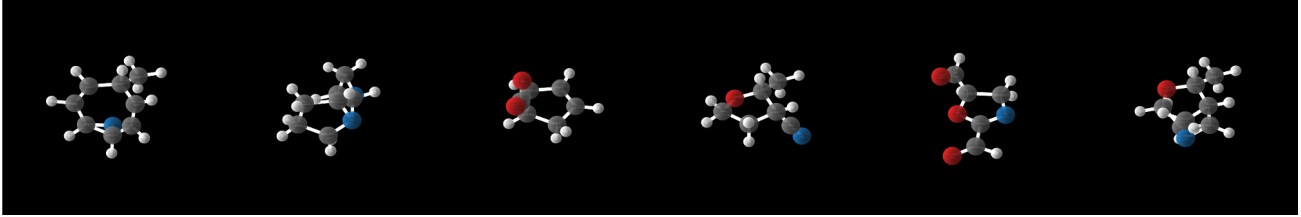

Figure 3: Molecule samples after applying the learned rotations.

## A. Visualization

We visualize the learned rotations in Figure 2 and Figure 3. We indeed found that after alignment, molecules tended to arrange common structural semantics (e.g., rings) in similar orientations.

