# OpenReview forum: "Scalable Non-Equivariant 3D Molecule Generation via Rotational Alignment"
_ICML.cc/2025/Conference — ICML 2025 poster_

### Official Review · Reviewer_bfzT · 2025-02-23

**Overall Recommendation:** 3

**Summary:**

This paper proposes Aligned Latent Diffusion Model (ALDM) to improve 3D molecule generation by relaxing SE(3) equivariance constraints in diffusion models. Instead of enforcing equivariance, ALDM learns a sample-dependent SO(3) transformation using an autoencoder to align molecular representations, enabling the use of non-equivariant diffusion models like vanilla GNNs and transformers. This approach significantly improves scalability and efficiency while maintaining state-of-the-art sample quality, outperforming previous non-equivariant models and matching equivariant baselines in molecule stability and validity.

**Claims And Evidence:**

The claims made in the submission are supported by clear and convincing evidence

**Essential References Not Discussed:**

NA

**Experimental Designs Or Analyses:**

Yes, The proposed method is evaluated on appropriate datasets with other baselines.

**Methods And Evaluation Criteria:**

The proposed method is evaluated on appropriate datasets with other baselines.

**Other Comments Or Suggestions:**

NA

**Other Strengths And Weaknesses:**

Strengths:
1. Improved Efficiency & Scalability – By removing SE(3) equivariance constraints, ALDM enables the use of simpler, more scalable architectures (e.g., GNNs, transformers), reducing computational costs and improving sampling speed.
2. Flexible Model Design – Unlike equivariant models that require specialized architectures (e.g., EGNNs), ALDM allows for standard non-equivariant GNNs and transformers, making implementation and optimization more accessible.
3. Well-Motivated Theoretical Insight – The paper challenges the necessity of equivariance in 3D molecule generation, providing a principled alignment-based approach that removes redundancy while maintaining symmetry-aware representations.

Weakness & Questions:
1.	The performance compromises. ALDM does not achieve the SOTA performance in each column on QM9 in Table 1.
2.	What’s the difference between Eq. 17 and regular message-passing GNNs? EGNN mainly convert positions to distance to achieve the equivariance. Without this design, Eq. 17 becomes a regular GNN.
3.	Alignment Quality Uncertainty – The autoencoder-based alignment is learned in an unsupervised manner, making it unclear whether the learned SO(3) transformations always provide an optimal latent representation for downstream diffusion modeling.
4.	Limited Theoretical Justification on Performance Trade-offs – While the paper argues against the necessity of equivariance, it does not provide a rigorous theoretical analysis explaining why non-equivariant models can fully match or surpass equivariant ones under all conditions.
5.	No code available. No appendix showing more information. E.g. the dataset information, the detailed experimental settings, the optimal hyperparameters….

**Questions For Authors:**

See Weaknesses & Questions

**Relation To Broader Scientific Literature:**

NA

**Theoretical Claims:**

Algorithms in this paper is correct.

---

> ### Author Rebuttal · Authors · 2025-03-31
>
> We thank Reviewer bfzT for taking the time to review our paper and provide valuable feedback. We respond to the concerns as follows.
>
> **Not SOTA on QM9**: Our paper mainly aims to improve non-equivariant diffusion models. On QM9, compared with the best non-equivariant baseline GraphLDM-aug, all three variants of our model improve the performance drastically, which validates the main claims of our paper. In fact, our best model ALDM$\_{\text{DiT-B}}$ achieves the highest Validity (94.2%) on QM9 across all (equivariant and non-equivariant) baselines and is only negligibly worse than GeoLDM on other metrics. We kindly request the reviewer to also consider the efficiency improvement shown in Table 3. Overall, our model achieves the best performance-efficiency trade-off on QM9. Additionally, on GEOM-DRUGS, which is much larger and more challenging than QM9, our model achieves the best performance.
>
> **Difference between Eq. 17 and regular GNN**: Eq. 17 is indeed a regular GNN. We use it as a basic non-equivariant architecture (for the decoder and the noise prediction network of the diffusion model). Unlike EGNN, it simply concatenates atom coordinates and types and is thus non-equivariant. In Table 1, both ALDM$\_{\text{GNN}}$ and the baseline GraphLDM (from the GeoLDM paper) use Eq.17 as the noise prediction network, except that learned rotations are applied to the latent space of ALDM$\_{\text{GNN}}$. The performance gain validates that the aligned latent space can improve the training of non-equivariant diffusion models.
>
> **Alignment Quality Uncertainty**: The lack of supervision is an inherent challenge for generative modeling. Our approach manages to learn rotations in such an unsupervised setting, which we believe is part of our contribution rather than a weakness of our model. Furthermore, the theme of our paper is to show that “alignment” is better than “no alignment” for non-equivariant diffusion models, which is already supported by our experiments. We agree with the reviewer that investigating the “optimality” of the aligned representations would be interesting, but we think it is currently beyond the scope of this paper. We’d love to explore this in future work.
>
> **Limited Theoretical Justification on Performance Trade-offs**: In theory [1], diffusion models are able to learn the dataset distribution as long as the score function is learned well, without imposing any constraints (e.g., equivariance) on the target distribution. Therefore, non-equivariant diffusion models, in principle, have the full potential to learn the dataset distribution. On the contrary, previous equivariant diffusion models took equivariance for granted and didn’t justify why equivariance is necessary.
> Additionally, a few recent works (e.g., [2] (AlphaFold 3), [3]) also demonstrate the superior performance of non-equivariant diffusion models, though the specific problems they address are different from ours.
>
> **Code and detailed experimental settings**: We already detailed the experimental setup in the main body of the paper (as mentioned by Reviewer t7B8: “The authors provided sufficient details about their experiment settings”). Specifically, we describe the dataset information in Section 5.1. We follow exactly the dataset preprocessing and splits of baselines EDM and GeoLDM and point to their public repositories. In Section 5.2, we introduce hyper-parameters (learning rate, number of layers, hidden dimension, batch size, etc.) we use before we present the experimental results. We will organize these information in a better format (e.g., using tables) in the next version. Our code will be released after the paper is accepted.
>
> [1] https://arxiv.org/abs/2011.13456 \
> [2] https://www.nature.com/articles/s41586-024-07487-w \
> [3] https://proceedings.mlr.press/v235/wang24q.html
>
> We kindly request Reviewer bfzT to consider raising their score if they think our response can address the concerns. We are more than happy to answer any further questions!

---

> > ### Comment · Reviewer_bfzT · 2025-04-05
> >
> > Thanks for the authors' responses. I have raised my score.

---

> > > ### Author Response · Authors · 2025-04-07
> > >
> > > We thank Reviewer bfzT for acknowledging our rebuttal and are pleased that our responses addressed the concerns.

---

### Official Review · Reviewer_Biv1 · 2025-03-10

**Overall Recommendation:** 4

**Summary:**

This paper proposes learning a roto-aligned latent space for 3d molecule generation. The alignment is achieved through learning sample-wise rotation with auto-encoder. More specifically, ALDM adopts an equivariant encoder and a non-equivariant decoder, largely alleviating the constraints in architecture design, making Transformers potential for large-scale pretraining of small molecules. Experiments on QM9 and GEOM-DRUG shows ALDM performs better than non-equivariant baseline, and on par with SOTA equivariant models.

**Claims And Evidence:**

Equivariance constraints may not be necessary for molecule generation.
Successfully supported.

**Essential References Not Discussed:**

There are some stronger baselines, yet the reviewer believes the results are enough to support main claims.

**Experimental Designs Or Analyses:**

1. Unconditional generation
Ablation of decoder choice is interesting, and it shows DiT's advantage over simple EGNN. Overall comparison is decent, ALDM achieves competitive results with GeoLDM.
2. Conditional generation
The results look normal. Even though property-conditioned generation is not so useful.

**Methods And Evaluation Criteria:**

Yes

**Other Comments Or Suggestions:**

No

**Other Strengths And Weaknesses:**

Strengths
1. The idea is simple, but elegant and effective, making non-equivariant Transformer more applicable to large scale pretraining of molecules. The scaling law and emergent ability of Transformer have already been proved in other domains.
2. Generally good results, making the claim convincing.
Weakness
1. Lack of stronger baselines.
2. Lack of analysis of learned alignment (the sample-wise rotation) and latent representation. The reviewer is quite curious about: 1. Does the learned rotation have some physical or chemical insights? Is there a natural canonical orientation for 3d molecules? 2. Does the VAE really learn chemical/structural semantics in latent space? More analysis or visualization would be insightful.

**Questions For Authors:**

See weakness.

**Relation To Broader Scientific Literature:**

N/A

**Theoretical Claims:**

N/A

---

> ### Author Rebuttal · Authors · 2025-04-01
>
> We thank Reviewer Biv1 for taking the time to review our paper and provide valuable feedback. We are glad to see that they found our approach interesting and our results good enough to support our claims. We respond to the concerns as follows:
>
> **Lack of stronger baselines**: We thank the reviewer for pointing this out! We will add more recent baselines (e.g., [1]) in an updated version of our paper. The performance of our non-equivariant models is still highly competitive compared with stronger equivariant baselines, and as the reviewer said, we believe the existing experimental results are sufficient to validate the effectiveness of our approach.
>
> [1] https://arxiv.org/abs/2403.15441
>
> **Lack of analysis of learned alignment and latent representation**.
> We provide more analysis below. First, for a single molecule, its canonical form can be obtained (e.g., by applying PCA to atom coordinates). However, the challenge is that such canonicalization may not have consistent effects for different molecules. For example, the directions (or signs) of the principal components obtained by PCA are ambiguous. Moreover, applying PCA to atom coordinates ignores the dependencies between coordinates and atom types, which should be generated together. Therefore, we believe it makes more sense to learn canonical forms across the training set by maximizing the overall reconstruction quality.
> As an ablation, we report the results based on PCA:
> PCA$\_{\text{GNN}}$ achieves 82.8% molecule stability on QM9 and 81.1% molecule stability on DRUGS. Its performance is better than the best non-equivariant baseline GraphLDM but still significantly worse than ALDM$\_{\text{GNN}}$ (87.4% on QM9 and 83.0% on DRUGS).
>
> To analyze the rotations learned by the autoencoder, we randomly sample a mini-batch of molecules from the training set and compare their raw positions and the positions after rotations. We provide some visualization in the following anonymous link:
>
> https://docs.google.com/document/d/e/2PACX-1vQeBYa4zljEoIM_h-4cvlOH-unSQFJhDgwVuq0FxtISIXZJ-T9T8q53OoWOKtoMR0d4O1SA7Vp12xq3/pub
>
> We indeed found that common structural semantics (e.g., rings) of some molecules are easier to recognize after the alignment.
> Since our encoder is equivariant, such rotations are applied to the latent representations in the same way and thus make them more applied
>
> We hope our response can address the reviewer's concerns. We are happy to answer any further questions.

---

> > ### Comment · Reviewer_Biv1 · 2025-04-03
> >
> > 1. In your response, I think 81.1% should atom stability on DRUG, right?
> > 2. It will be helpful if you can extract some molecule-level latent representations and conduct analysis/visualization based on that.
> > 3. What is PCA_{GNN}? Why is it so good? Actually I'm interested in your learned representations instead of coordinates.
> > 4. What do you mean by "rings are easier to recognize after alignment"? I can see that, after alignment, the molecules tend to arrange the ring along a very similar central axis, i.e., perpendicular to the plane of the paper. Does this reflect some natural canonical orientation on most of the molecules? Why does the model learn such kind of orientation?

---

> > > ### Author Response · Authors · 2025-04-03
> > >
> > > We thank Reviewer Biv1 for reading our rebuttal.
> > >
> > > 1. Yes. Thanks for catching the typo.
> > > 2. We add the visualization of latent representations in the following link:
> > >
> > > https://docs.google.com/document/d/e/2PACX-1vQIraK3xLJadm1U3iONPV4u6ZW52EbTofu4027WMFAhJylIplEZhv-BkgkJZWMu99yb75F88yrCAfgF/pub
> > >
> > > The first row shows molecule samples from the training set. The second row applies the learned rotations to the original molecules. The third row exhibits the latent representations generated by the encoder for corresponding molecules.
> > >
> > > Specifically, the encoder (an EGNN) generates a 4-dimensional latent representation for each atom, and the molecule-level latent representation is a set of atom representations. The first 3 dimensions of the atom latent representation are equivariant features (i.e., the $\mathbf{x}$ part of Eq (15)), and the last dimension encodes the invariant feature (i.e., the $\mathbf{h}$ part of Eq (15)). To plot latent representations, we treat the first 3 dimensions as coordinates and map the continuous values of the last dimension to colors (using colormaps of Matplotlib). As we can observe from the visualization, the latent representations very well preserve the relative positions between atoms, and the colors are well separable (indicating that atom types can be easily decoded).
> > >
> > > 3. PCA$\_{\text{GNN}}$ is a baseline that applies preprocessing (i.e., PCA) to find a canonical form of a molecule. PCA finds the three directions with the largest variances in atom coordinates (via calculating eigenvectors of the covariance matrix) and makes them the new axes. PCA$\_{\text{GNN}}$ then runs a non-equivariant diffusion model (with a basic GNN denoising network Eq (17)) over the new coordinates. The good performance of PCA$\_{\text{GNN}}$ supports the benefit of alignment. However, as we explained in the rebuttal, the directions of the axes found by PCA are ambiguous (because eigenvectors are still valid if signs are flipped), which can make the canonicalization inconsistent. Furthermore, PCA ignores the dependencies between atom coordinates and other atom features (e.g., types) that should be generated together. The performance of PCA$\_{\text{GNN}}$ is significantly worse than our model ALDM$\_{\text{GNN}}$, which further supports our approach that uses neural networks to learn alignment.
> > >
> > > 4. By "rings are easier to recognize after alignment" we meant that after alignment molecules tend to arrange rings in similar orientations. We speculate that the model learns such orientation because a large fraction of molecules with rings in the training set already have (or are not far away from) this orientation (e.g., the 3rd, 4th and 5th of the visualization), so it is easier for the model to align the remaining molecules with this orientation. In general, we believe the specific canonical form learned by the model depends on the dataset and the stochastic training process and may not be pre-determined.

---

### Official Review · Reviewer_uMCt · 2025-03-13

**Overall Recommendation:** 3

**Summary:**

The paper explores a non-equivariant alternative to protein generation. It uses an explicit rotation network to rotate zero-centered molecule coordinates and builds a latent space on top of the rotated coordinates. The latent space is learned via VAE objective and encodes aligned features. This allows one to learn a non-equivariant diffusion model on the latent space, as motivated by those used in aligned 3D point cloud literature. The method achieves comparable performance as equivariant counterparts for generation while achieving faster sampling speed.

## Update after rebuttal

The authors have adequately addressed my concerns regarding the non-equivariant decoder and scaling experiments. I increased score accordingly.

**Claims And Evidence:**

- The authors also claim that the non-equivariant diffusion model allows more "scalability" while no experiments are done to validate that this model "scales".

- The authors claim that non-equivariant decoder is necessary to learn the aligned latent representation. Why is this the case? The paper does not further elucidate the intuition and does not design experiments to validate the claim.

- The design of the encoder makes it such that the latent representation is supposedly aligned, which is the core of the method. However, the authors do not conduct experiments on investigating whether the learned latent spaces are actually aligned. Some simple PCA or other analysis can be done to investigate it further.

**Essential References Not Discussed:**

Related works are properly discussed.

**Experimental Designs Or Analyses:**

Pros:
- The paper validates that the non-equivariant diffusion is faster due to parallelizable transformer network by showing sampling wall-clock time. It similarly shows by experiments that the generation performance is comparable.


Cons:
- One simple baseline the paper is missing is simply to train fully non-equivariant autoencoder and non-equivariant latent diffusion with the same architecture. This is to validate the claim that the supposedly aligned latent feature and the newly introduced rotation network is actually useful. it is not clear whether the retained performance comes from the aligned latent space or the new transformer itself.
- Following the above point, it is also not clear whether the latent space is actually aligned or could it be optimized away by some non-equivariant encoders.

**Methods And Evaluation Criteria:**

The paper mostly follows previous papers in terms of datasets and in terms of evaluating generation quality.

**Other Comments Or Suggestions:**

As mentioned above.

**Other Strengths And Weaknesses:**

Strength:
- The writing is very clear and easy to follow.

Weaknesses:
- The proposal is incremental and novelty is limited.
- The results show marginal improvement in terms of quality and some improvement in sampling speed only due to a change of architecture. The results do not strongly justify the reason for adopting this method.
- The paper claims aligned feature helps, but does not actually conduct experiments to check that the latent space is aligned and the improvements are due to this design rather than other factors such as changing to a different transformer architecture.

**Questions For Authors:**

I have listed all my concerns in previous sections and wish the authors to address. I can consider increasing score if they are properly addressed.

**Relation To Broader Scientific Literature:**

The paper aims to advance core algorithms of drug generation and increase efficiency of generation while retaining quality.

**Theoretical Claims:**

The method does not contain theories.

---

> ### Author Rebuttal · Authors · 2025-04-01
>
> We thank Reviewer uMCt for taking the time to review our paper and provide valuable feedback. We respond to the concerns as follows:
>
> **Scalability of non-equivariant diffusion models**: In fact, we did experiments to show the scalability of our model. We kindly refer the reviewer to Table 3 on page 8, where we show the model sizes. We tested non-equivariant denoising networks of three different sizes for the diffusion model: GNN is from Eq (17), DiT-S is the small version model from the diffusion transformer paper, and DiT-B is the base version with a larger hidden dimension (4 times \#params as DiT-S and more than 10 times \#params as GeoLDM). Our largest model achieves the best performance on the DRUGS dataset and still maintains higher training/sampling efficiency than existing equivariant diffusion models.
>
> **Why is non-equivariant decoder necessary to learn alignment?**: Let $x$ denote the original atom coordinates and $x' = \mathcal{D}(\mathcal{E}(x))$ denote the reconstructed coordinates. The reconstruction error reduces to L2 loss $\Vert x' - x \Vert^2$ for continuous coordinates. Now suppose $x$ is rotated by $R$, i.e., $Rx$. If both the encoder and decoder are equivariant, the reconstruction would become $Rx'$, and the L2 loss between $Rx$ and $Rx'$ would be the same as $\Vert x' - x\Vert^2$ (because $R$ is orthogonal), making the rotation network unable to receive any supervision signals. Therefore, encoder and decoder should not both be equivariant.
>
> We chose to make the decoder non-equivariant and keep the encoder equivariant because we wanted to compare directly with the baseline GeoLDM. GeoLDM uses equivariant networks for both encoder and decoder. We use the same equivariant encoder as GeoLDM to keep the architectural factors affecting the generation of latent representations the same, except that our model applies the learned rotations.
>
> **Ablations on the architecture and latent space**. Our experiments already contain ablations. We kindly refer the reviewer to the performance of GraphLDM and ALDM$\_{\text{GNN}}$ in Table 1. GraphLDM is a non-equivariant diffusion model baseline from the GeoLDM paper that replaces the original equivariant denoising network of the diffusion model with a non-equivariant network (Eq 17). ALDM$\_{\text{GNN}}$ also uses Eq (17) as the denoising network for diffusion and the same hyper-parameters (noise schedule, diffusion steps, etc.). Therefore, the difference between ALDM$\_{\text{GNN}}$ and GraphLDM lies in that the latent space of ALDM$\_{\text{GNN}}$ is aligned by rotations. The experimental results show a significant performance gain of ALDM$\_{\text{GNN}}$ compared with GraphLDM, which validates that the aligned representations improve the learning of non-equivariant diffusion models. We also tried to use DiT for GraphLDM, but the gap was still significant.
>
> Additionally, we randomly sample a mini-batch of molecules from the training set and compare their original and rotated positions. The visualization is contained in the following anonymous link:
> https://docs.google.com/document/d/e/2PACX-1vQeBYa4zljEoIM_h-4cvlOH-unSQFJhDgwVuq0FxtISIXZJ-T9T8q53OoWOKtoMR0d4O1SA7Vp12xq3/pub
> We indeed found that common structural semantics (e.g., rings) are easier to recognize after rotation. Since our encoder is equivariant, these rotations are applied to the latent representations in the same way and thus make them more aligned.
>
> **Novelty of the proposed method**.
> While the evaluation of novelty depends on one's own perspective, we want to emphasize that our paper is not simply “a change of architecture”. In fact, in this paper, we switch from equivariant models to non-equivariant models and propose an approach to close the gap between them. As we show in the experiments (explained above), the improvement brought by our approach is significant even if we use the same non-equivariant architecture (i.e., a basic GNN) for the diffusion model.
>
> Besides, our model not only improves the sampling speed, but also significantly saves training cost. On QM9, training the baseline equivariant diffusion model (e.g., GeoLDM) to convergence takes nearly 3000 epochs, which needs 4 days to finish on a single 4090 GPU (in Table 3 on page 8 we show the average training time per epoch). As a comparison, the training of our largest model ALDM$\_{\text{DiT-B}}$ takes 3 days on a 4090 GPU. We use a larger batch size and thus more epochs for ALDM$\_{\text{DiT-B}}$, but the total training cost is still reduced significantly. We believe this provides more support for our model.
>
> We hope our response can provide a clearer explanation of the key points of our work and kindly request Reviewer uMCt to consider revising their score if they think our response can address their concerns. We are happy to answer any further questions.

---

### Official Review · Reviewer_t7B8 · 2025-03-13

**Overall Recommendation:** 3

**Summary:**

The paper suggests learning a sample-dependent rotational transformation during molecule generation. This approach aligns the molecules to specific directions, eliminating the necessity of employing equivariant models. The proposed method demonstrates promising performance and efficiency on benchmark datasets.

**Claims And Evidence:**

The claims presented in this paper are substantiated by references and detailed experiments.

**Essential References Not Discussed:**

The paper lacks references of the research on canonicalization. This should encompass, but is not limited to, the following:
- Ma, G., Wang, Y., Lim, D., Jegelka, S., & Wang, Y. A Canonicalization Perspective on Invariant and Equivariant Learning. In The Thirty-eighth Annual Conference on Neural Information Processing Systems.
- Tahmasebi, B., & Jegelka, S. Generalization Bounds for Canonicalization: A Comparative Study with Group Averaging. In The Thirteenth International Conference on Learning Representations.
- Dym, N., Lawrence, H., & Siegel, J. W. (2024, July). Equivariant Frames and the Impossibility of Continuous Canonicalization. In International Conference on Machine Learning (pp. 12228-12267). PMLR.
- Ma, G., Wang, Y., & Wang, Y. (2023). Laplacian canonization: A Minimalist Approach to Sign and Basis Invariant Spectral Embedding. Advances in Neural Information Processing Systems, 36, 11296-11337.
- Kaba, S. O., Mondal, A. K., Zhang, Y., Bengio, Y., & Ravanbakhsh, S. (2023, July). Equivariance with learned canonicalization functions. In International Conference on Machine Learning (pp. 15546-15566). PMLR.

**Experimental Designs Or Analyses:**

The authors provided sufficient details about their experiment settings, and the performance and efficiency of their method on real-world tasks seem promising. However, as mentioned, this paper lacks several ablation experiments.

1. Comparing the use of a non-equivariant encoder or an equivariant decoder would help justify the authors’ architectural decisions.

2. Comparing their method with an equivariant canonicalization network or a canonicalization algorithm which ensure the rotational invariance of the aligned molecules.

3. Additionally, the authors claim that the aligned latent space reduces variations caused by Euclidean symmetries, but this assertion lacks empirical evidence. The authors could validate this claim through experiments, demonstrating a smaller variance in the latent space compared to non-equivariant baselines.

**Methods And Evaluation Criteria:**

First and foremost, I would like to emphasize that the authors’ concept of “aligning” molecules is known as *canonicalization* [1], and the resulting molecules are referred to as “canonical forms”. The authors inadvertently overlooked an extensive body of research on canonicalization. Notably, their method of learning a rotational matrix using a network is a weaker version of *learned canonicalization* [2,3], where the authors do not strictly enforce rotational equivariance on the canonicalization network. Since the authors’ approach is essentially canonicalization, it inherits the drawbacks of canonicalization methods. For instance, it imposes additional computational burden on the network to learn canonicalization and is unlikely to be more sample-efficient than equivariant networks. Furthermore, it lacks smoothness, meaning that a slight perturbation of the input molecule can result in a significant change in its canonical form.

The authors propose employing a non-equivariant network to learn the rotation matrix, thereby the learned “canonical form” $\mathbf{R}\_\theta\mathbf{x}$ is not guaranteed to be invariant to rotations. In contrast, the authors could consider using an equivariant network for this purpose as well, which would guarantee that $\mathbf{R}_\theta\mathbf{x}$ remains strictly invariant. However, using an equivariant network may be computationally inefficient, so the authors should also explore the use of a deterministic canonicalization algorithm. For instance, the simplest approach to “align” a molecule in 3D space would be applying Principal Component Analysis (PCA). This method involves identifying the top 3 directions with the highest variance in atom positions and aligning them with the standard basis (the $x,y,z$ axes). This method is even more efficient than the authors’ approach.

Furthermore, the authors assert that equivariant models are not essential for molecule generation since we only care about the overall probability of all possible positions. This argument holds true. However, in Section 4.1, it appears that the authors are still using an equivariant network for the encoder. The authors should provide justification for this choice through ablation experiments. Additionally, if we employ an equivariant canonicalization network or a deterministic canonicalization algorithm, the use of an equivariant encoder becomes unnecessary, as the aligned molecules will be guaranteed to be invariant.

---
[1] Ma, G., Wang, Y., Lim, D., Jegelka, S., & Wang, Y. A Canonicalization Perspective on Invariant and Equivariant Learning. In The Thirty-eighth Annual Conference on Neural Information Processing Systems.

[2] Kaba, S. O., Mondal, A. K., Zhang, Y., Bengio, Y., & Ravanbakhsh, S. (2023, July). Equivariance with learned canonicalization functions. In International Conference on Machine Learning (pp. 15546-15566). PMLR.

[3] Sareen, K., Levy, D., Mondal, A. K., Kaba, S. O., Akhound-Sadegh, T., & Ravanbakhsh, S. (2025). Symmetry-Aware Generative Modeling through Learned Canonicalization. arXiv preprint arXiv:2501.07773.

---

**Update: I had previously overlooked that [3] is a concurrent work released on January 15 of this year. In light of this, I have updated my score accordingly.**

**I would like to clarify my earlier rebuttal comment. When I stated that the paper "lacks empirical evidence demonstrating that canonicalization reduces the variance of latent representations across different input orientations," I meant that the authors should explicitly measure and report the variance of the learned latent representations, and compare this to appropriate baselines. This would provide stronger support than the current anecdotal examples where certain molecular ring structures appear similarly aligned. Such qualitative examples are insufficient to substantiate the claim that performance gains arise from improved alignment in latent space due to canonicalization.**

**I appreciate the additional ablation studies provided in the rebuttal and follow-up comment, and I recommend that the authors include these results in the paper, as they are essential for its completeness. Additionally, prior work on canonicalization should be acknowledged and discussed in the camera-ready version, which may require substantial revisions to the current writing.**

**Other Comments Or Suggestions:**

It would be more readable to bold the best performance in Tables 1 and 2.

**Other Strengths And Weaknesses:**

The strengths and weaknesses are adequately discussed in the preceding points.

**Questions For Authors:**

I have no additional questions.

**Relation To Broader Scientific Literature:**

The proposed method enhances the performance and efficiency of 3D molecule generation, benefiting the broader AI for science community.

**Theoretical Claims:**

There are no theoretical claims in this paper.

---

> ### Author Rebuttal · Authors · 2025-03-31
>
> We thank Reviewer t7B8 for taking the time to review our paper and provide valuable feedback. We respond to the concerns as follows.
>
> **Relation to canonicalization**: We thank the reviewer for pointing this out! We admit that we overlooked the literature on canonicalization, and we will add them to the related work. Furthermore, we want to highlight some differences between our paper and existing works on canonicalization:
> 1. Existing canonicalization works mostly consider the supervised learning setting, while our paper studies a generative modeling problem.
> 2. The main computational cost of equivariant baselines is the training of diffusion model (e.g., ~3000 epochs on QM9). Our approach uses an autoencoder to learn canonicalization in a few epochs before training the diffusion model. The additional overhead from learning canonicalization is marginal compared with the great speedup of training diffusion models (Table 3).
> 3. While theoretical works have analyzed the sample complexity gain of equivariance under the supervised learning setting, their conclusions may not be trivially extendable to generative modeling. Empirically, recent works (e.g., notable AlphaFold 3) also show the good performance of non-equivariant generative models. We believe this problem is worth further investigation, and our work is one step in this direction.
> 4. Regarding the smoothness of canonicalization, the rotation estimation by Eq (16) is smooth except when det(M)=0, which doesn’t notably impede training in practice [1]. Besides, we conjecture that the discontinuity issue of canonicalization is less of a concern in our setting, because we don't test canonicalization on unseen data and only use it to reshape the training data of which we have full control.
>
> **Ablations on architectural choices**: Our experiments already contain ablations. Specifically, our baseline GeoLDM uses equivariant networks for both encoder and decoder. We use the same equivariant encoder as GeoLDM to keep the architectural factors affecting the generation of latent representations the same, except that our model applies the learned rotations. Next, to learn rotations, our decoder has to be non-equivariant; otherwise, the reconstruction loss for atom coordinates (L2 loss) would be invariant to rotations and prevent the rotation network from learning anything. Note that although we have an equivariant encoder, it is only used to generate latent representations, and the probability distribution is learned by a non-equivariant diffusion model. This doesn't violate the claim of the paper.
>
> For the ablation results, we kindly refer the reviewer to the performance of GraphLDM and ALDM$\_{\text{GNN}}$ in Table 1. GraphLDM is a non-equivariant baseline from the GeoLDM paper that replaces the original equivariant denoising network of the diffusion model with a non-equivariant network (Eq 17). ALDM$\_{\text{GNN}}$ also uses Eq (17) as the denoising network for diffusion and the same hyper-parameters (noise schedule, diffusion steps, etc.). As a result, the significant performance gain of ALDM$\_{\text{GNN}}$ comes from the rotations applied to the latent space, which validates the effectiveness of our proposal. We also tried to use DiT for GraphLDM, but the gap was still significant.
>
> **Use of a canonicalization algorithm (PCA)**: We considered PCA in our preliminary experiments. There are two issues with PCA. First, the directions (or signs) of the principal axes are ambiguous. For prediction tasks, frame averaging can address this ambiguity, but it's not applicable to generative modeling. Second, applying PCA to atom coordinates ignores the dependencies between coordinates and atom types, which should be generated together. To further address the reviewer’s concern, we report here the results based on PCA:
> PCA$\_{\text{GNN}}$ achieves 82.8% molecule stability on QM9 and 81.1% molecule stability on DRUGS. Its performance is better than the best non-equivariant baseline GraphLDM but still significantly worse than ALDM$\_{\text{GNN}}$ (87.4% on QM9 and 83.0% on DRUGS)
>
> **Use of an equivariant canonicalization network**: We agree that using an equivariant canonicalization network (the rotation network in the paper) makes the canonical form of a molecule rotation invariant. However, to learn the specific canonical forms for different molecules that facilitate the reconstruction of the training set, the canonicalization network still needs to be trained jointly with the VAE (and probably less efficient). We'll try it in the updated version. Since our paper mainly aims to improve non-equivariant diffusion models, we believe the existing results are sufficient to support our claim.
>
> [1] https://arxiv.org/abs/2006.14616
>
> We hope our response can provide a clearer explanation of the key points of our paper and kindly request Reviewer t7B8 to consider revising their score if they think our response can address their concerns. We are happy to answer any further questions.

---

> > ### Comment · Reviewer_t7B8 · 2025-04-04
> >
> > Thank you for the detailed response. However, several of my concerns remain unaddressed. For instance, the proposed method appears to be a special case of learned canonicalization, and the relationship between these two methods is not adequately discussed. Furthermore, the authors assert that the performance gain of their method stems from aligning molecules to a canonicalized orientation, thereby reducing the variance in the latent space. However, this claim lacks sufficient evidence. The authors failed to provide empirical evidence demonstrating that canonicalization reduces the variance of latent representations across different input orientations. Additionally, they did not compare their approach to existing canonicalization methods that are exactly rotation invariant. Moreover, the authors neglected to consider previous works on canonicalization, resulting in an unorganized presentation of the writing and experiments. Therefore, I am inclined to retain my current score. I encourage the authors to address these concerns in future revisions of the paper.

---

> > > ### Author Response · Authors · 2025-04-07
> > >
> > > We thank Reviewer t7B8 for their comment. We would like to point out that, in our first-round rebuttal, we have already responded to the concerns restated in the reviewer's new comment. We expand more in the following.
> > >
> > > > For instance, the proposed method appears to be a special case of learned canonicalization, and the relationship between these two methods is not adequately discussed.
> > >
> > > As we explained in the rebuttal ("Relation to canonicalization"), the biggest difference between our paper and existing works on canonicalization is that we study canonicalization in the context of diffusion models (generative modeling), while existing canonicalization works focus on supervised learning. To be more specific:
> > > - We learn canonical forms of molecules in an unsupervised manner (through a non-equivariant VAE), without access to ground truth labels.
> > > - We utilize the aligned latent representations learned by the VAE to train non-equivariant diffusion models, yielding sample quality on par with equivariant diffusion models and significantly higher efficiency.
> > >
> > > While we agree with the reviewer that our method is related to existing works on learned canonicalization and we are happy to add them to related work, we believe that given the above differences, our work should not be simply viewed as "a special case" of learned canonicalization.
> > >
> > > The only exception that considers canonicalization and diffusion models is [1], which is a workshop paper that became online after 14 Jan 2025, so we believe it belongs to concurrent work. Notably, our method is also different from [1]. [1] merges an equivariant canonicalization network with the denoising network of the diffusion model and therefore inherits the inefficiency of training equivariant diffusion models. In contrast, our method learns canonicalization using a lightweight non-equivariant VAE and then trains non-equivariant diffusion models in the aligned latent space. The performance of [1] (84.6% molecule stability on QM9) is significantly worse than ours (87.4%) with the same diffusion backbone.
> > >
> > > [1] https://arxiv.org/abs/2501.07773
> > >
> > >
> > > > Furthermore, the authors assert that the performance gain of their method stems from aligning molecules to a canonicalized orientation, thereby reducing the variance in the latent space. However, this claim lacks sufficient evidence. The authors failed to provide empirical evidence demonstrating that canonicalization reduces the variance of latent representations across different input orientations
> > >
> > > In fact, we already provided empirical evidence to support this claim:
> > > - We visualize the learned rotations and latent representations in the following link (due to character limit, the link was put in our responses to Reviewer uMCt and Reviewer Biv1, and we paste it here): https://docs.google.com/document/d/e/2PACX-1vQIraK3xLJadm1U3iONPV4u6ZW52EbTofu4027WMFAhJylIplEZhv-BkgkJZWMu99yb75F88yrCAfgF/pub
> > > We indeed observe that the learned rotations enable latent representations to arrange common structural semantics (e.g., rings) in similar orientations.
> > > - Besides, as we noted in our rebuttal, our experiments already included ablations (e.g., ALDM$\_{\text{GNN}}$ vs GraphLDM) that validated the significant improvement resulted from the aligned latent space, under the same encoder architecture and same diffusion backbone.
> > >
> > > > Additionally, they did not compare their approach to existing canonicalization methods that are exactly rotation invariant
> > >
> > > As we explained in our rebuttal, replacing our rotation network (a regular GNN) with an equivariant neural network is just a variation of our method and won't affect the claim or contributions of our paper.
> > >
> > > To further address the reviewer's concern, we experiment with it and report the result here: Specifically, we replace our rotation network with an EGNN and let it generate the rotation matrix row by row. This makes the canonical form of a molecule rotation-invariant as the reviewer suggested. Under the same non-equivariant diffusion model (with a regular GNN (Eq 17) as the denoising network), the performance of this method (*98.6% atom stability and 87.8% molecule stability* on QM9) is negligibly different from our current performance (*98.7% atom stability and 87.4% molecule stability*).
> > >
> > > > Moreover, the authors neglected to consider previous works on canonicalization, resulting in an unorganized presentation of the writing and experiments
> > >
> > > While we understand the reviewer's concerns and acknowledge that our submission in its initial status was not comprehensive enough, we sincerely request the reviewer to consider our rebuttal, in which we responded to every raised concern, as a complement and improvement to our initial submission. We hope this can change the reviewer's initial impression.

---

### Decision · Program_Chairs · 2025-05-01

**Decision:**

Accept (poster)

**Comment:**

This paper presents an approach to unlock scalability in 3D molecule generative models by relaxing the equivariant constraints in previous architectures. The proposed approach first learns a sample dependent SO(3) transformation and then a generative model is trained in this aligned space. The paper is well written and the experimental results support the claims made in the paper. Reviewers have pointed out the simplicity of the approach and the elegant formulation of the problems as strengths. The AC agrees that this is paper is a valuable contribution for ICML.